# Cheese Intake Exhibits an Alteration of Glycolipid Profile and Impacts on Non-Alcoholic Fatty Liver in Bahraini Older Adults

**DOI:** 10.3390/geriatrics7040075

**Published:** 2022-07-20

**Authors:** Simone Perna, Layla H. Hammad, Mohamed Wael Mohamed, Dalal Alromaihi, Mariam Alhammadi, Noora Al-Khater, Anas Rashed Alchuban, Mawadh Ali Aledrisy, Zahra Ilyas, Tariq A. Alalwan, Mariangela Rondanelli

**Affiliations:** 1Department of Biology, College of Science, University of Bahrain, Sakhir P.O. Box 32038, Bahrain; zahra.muhammadilyas@gmail.com (Z.I.); talalwan@uob.edu.bh (T.A.A.); 2Ministry of Health, Manama P.O. Box 32038, Bahrain; laylabh2018@outlook.com; 3Internal Medicine Department, King Hamad University Hospital, Muharraq P.O. Box 32038, Bahrain; mohdwael.93@hotmail.com (M.W.M.); dr.alromaihi@gmail.com (D.A.); mariamhammadi07@gmail.com (M.A.); noora.alkhater@hotmail.com (N.A.-K.); dr.anas90@icloud.com (A.R.A.); 4Royal College of Surgeons in Ireland, Medical University of Bahrain, Adliya P.O. Box 15503, Bahrain; 5Arabian Gulf University, Manama P.O. Box 32038, Bahrain; mahmaa21@gmail.com; 6IRCCS Mondino Foundation, 27100 Pavia, Italy; mariangela.rondanelli@unipv.it; 7Unit of Human and Clinical Nutrition, Department of Public Health, Experimental and Forensic Medicine, University of Pavia, 27100 Pavia, Italy

**Keywords:** metabolic syndrome, older adults, dietary pattern, food consumption

## Abstract

***Background***: Metabolic syndrome (MetS) is a multifactorial condition characterized by a complex interrelation between genetic and environmental factors that heighten the risk of cardiovascular diseases and all-cause mortality. It is hypothesized that diet may play an important role in the regulation of metabolic syndrome factors and influence the process. Therefore, the main objective of this study was to investigate the specific dietary patterns associated with metabolic syndrome markers and quantify the possible effects of dietary patterns among Bahrain older adults. ***Methods***: This is a cross-sectional study that included 151 Bahraini patients diagnosed with MetS, 89 (58.7%) were females and 62 (41.3%) males. ***Results***: The prevalence of Non-Alcoholic Fatty Liver was 89%. Statistically significant correlations were found between dairy products with low fat and SBP (r = 0.182, *p* < 0.001) body mass index (BMI) (r = −0.195; *p* < −0.01). Higher chicken consumption was associated with reduction of BMI (r = −0.273; *p* < −0.01). A higher consumption of ricotta and cheddar cheese (high in fat) was associated with higher levels of triglycerides (*p* < 0.01). Higher frequent consumption of rice (basmati) was associated with lower glucose levels (r = −0.200; *p* < −0.01). Fatty liver has been associated with high consumption of cream cheese (*p* < 0.01). ***Conclusion***: In older Bahraini adults with metabolic syndrome, higher frequency of food consumption of full-fat cheese was linked with a derangement of lipid profile and Non-Alcoholic Fatty Liver. Positive effects on BMI have been recorded with higher-frequency consumption of basmati rice and chicken.

## 1. Introduction

Metabolic syndrome (MetS) is defined as a multifactorial condition characterized by a complex interrelation between genetic and environmental factors that higher the risk of diseases, including cardiovascular diseases (CVD) and all-cause mortality [1]. In recent decades, the prevalence of the MetS phenomenon has affected all age and social groups. Kaur [1] reported that the prevalence of MetS was between 10% and 84%. In Bahrain, Khalil et al. [2] studied the prevalence of MetS among patients with bipolar affective disorder and found that its prevalence to be 31.84% in the control group general population. 

Several studies shed in light that refined carbohydrates, including syrups, biscuits, and cakes (very common in the Middle East) are closely associated with MetS, type 2 diabetes (T2D), and cardiovascular diseases (CVD). In addition, due to the contamination from the western diet and the phasing out of the traditional diet, in the entire Middle East, the majority of foods are rich in saturated fats and trans fatty acids. Specifically, saturated fat is rising more than polyunsaturated or monounsaturated fats, with adverse effects on insulin sensitivity [3,4].

An unhealthy diet has been implicated as a risk factor for the development of chronic diseases related to MetS, such as hypertension, impaired fasting plasma glucose (FPG), dyslipidemia, and obesity [5]. Moreover, the nutritional quality of dietary patterns plays a leading role in MetS development, with a western dietary pattern (e.g., high in red meat, fried foods and refined carbohydrates and sugars) being associated with higher risk of MetS and CVD [6]. For instance, Naja et al. [7] reported that unhealthy dietary patterns, focusing on animal proteins and processed foods with higher energy, fat and sugar intakes was associated with MetS in Lebanese adults. Furthermore, Al-Daghri et al. [8], studying Saudi adults, found that lower intakes of potassium, calcium, zinc, vitamin A, vitamin E, or vitamin C were associated with higher prevalence of MetS.

Adherence to healthful dietary patterns, which include nutrient-dense fruits, vegetables, whole grains, nuts and legumes, and low intakes of refined carbohydrates, and animal products are inversely associated with reduced prevalence of MetS [9]. Wei et al. [10], in their study on a Chinese population, showed that the traditional Chinese dietary pattern (high in vegetables, fruits, whole grains, soyabean) correlated negatively with MetS risk factors, while the animal food pattern (high in meats and seafood) showed a positive association with MetS risk factors.

A recent meta-analysis by Neuenschwander showed that there was no association between total fat intake and the incidence of T2D. In particular, a high intake of vegetable fat was inversely associated with T2D incidence [11].

Another study by Zeraatkar showed low-certainty evidence that a reduction in processed meat intake to three servings per week was associated with a very slightly lower risk for all-cause mortality, cardiovascular mortality and stroke [12].

The aim of this study is to identify whether there are associations between different MetS components and non-alcoholic fatty liver and specific food intake in a sample of Bahraini adults aged between 50 and 90 years.

Based on these observations, it is hypothesized that diet may play an important role in the regulation of metabolic syndrome factors and influence the process. Therefore, the main objective of this study was to investigate the specific dietary patterns associated with metabolic syndrome markers and quantify the possible effects of dietary patterns among Bahraini adults. The results of this study can provide a reference for future dietary evaluation and dietary intervention in metabolic syndrome management in Bahrain.

## 2. Methods and Materials

### 2.1. Study Design

This is a cross-sectional study conducted among the Bahraini population. Data collection was performed during the period from May to July 2020. All participants gave their informed consent prior to their inclusion in the study. The study was conducted in accordance with the Declaration of Helsinki, and the protocol was approved by the Institutional Research Board at King Hamad University Hospital (KHUH), Bahrain and the Ethics Committee of the Department of Biology, University of Bahrain, Bahrain (protocol n. 20-323).

### 2.2. Sample Size and Inclusion and Exclusion Criteria

The sample size was representative of the population with a confidence level of 80% and a confidence interval of 5%. The sample size was determined using an online sample size calculator (survey system) and using a specific equation *n* = N x/((N − 1)E2 + x), and the estimated sample size was 190 subjects in accordance with the study by Alzeidan et al. [13].

The study sample was randomly selected from a section of the population by going through the medical records of MetS patients who visited the endocrinology outpatient clinic of KHUH. The inclusion criteria were age 50 years and above, of Bahraini nationality, and not diagnosed with any psychiatric disorder. Participants were excluded if they were under 50 years of age, non-Bahraini nationals, or diagnosed with any psychiatric conditions. A total of 151 patients were included in the study, with 74 participants completing the online questionnaire. The sample size was calculated using online tools (https://clincalc.com/stats/samplesize.aspx, accessed on 1 April 2019).

### 2.3. Data Collection

Data pertaining to biochemical parameters were utilized from the electronic medical records system (Health Outcomes and Patient Experience; HOPE) of KHUH.

Fasting venous blood samples were drawn with the participants seated. Blood was collected and handled under strictly standardized conditions. Blood samples were collected into vacuum tubes without anticoagulant, left for 1 h at room temperature, and then centrifuged for 15 min at 1500× *g* at 20 °C. The serum was then transferred into plastic tubes, rapidly frozen, and stored at –80 °C until analysis (<1 mo later). Whole blood (with the use of EDTA as an anticoagulant) was used for hematologic variables. Clinical chemistry markers were detected on the Roche Cobas Integra 400 plus analyzer (Roche Diagnostics, Basel, Switzerland), with especially designed commercial kits provided by the manufacturer. Cobas Integra 400 is a random, continuous-access, sample-selective analyzer that provides absorbance photometry for measuring enzymes and substrates, turbidimetry for specific proteins, and ion-selective electrode potentiometry for serum electrolytes. Cholesterol, triglycerides (TG), high-density lipoprotein-cholesterol (HDL-C), total proteins, total bilirubin, iron, glucose, uric acid, creatinine, and liver enzymes such as alanine transaminase, aspartate transaminase, and γ-glutamyltransferase were measured by enzymatic-colorimetric methods. C-reactive protein (CRP) was determined by a nephelometric high-sensitivity CRP (Dade Behring, Deerfield, IL, USA).

Erythrocyte, white blood cell, and platelet counts; hemoglobin concentrations; mean cell volumes; and mean cell hemoglobin concentrations were measured with the use of a Coulter automated cell counter (MAX-M; Beckman Coulter, Brea, CA, USA). Serum albumin was analyzed with the use of a nephelometric method (Behring Nephelometric Analyzer II, Behring Diagnostics), with a 2% CV.

Additional data variables needed to categorize patients were collected. Participants who met the National Institutes of Health guidelines for MetS components, with three or more of the following five conditions were defined as having MetS [14]: central obesity (≥102 cm for men and ≥88 cm for women), elevated blood pressure (systolic blood pres-sure; SBP ≥130 mm Hg and/or diastolic blood pressure; DBP ≥85 mm Hg), reduced HDL-C level (<40 mg/dL for men and <50 mg/dL for women), TG (≥150 mg/dL), and elevated FPG levels (≥100 mg/dL).Due to the coronavirus (COVID-19) pandemic, categorized MetS patients were contacted by phone using a standardized verbal script to assure informed consent in filling the electronic questionnaire entitled “Dietary and lifestyle patterns in metabolic syndrome patients in the Kingdom of Bahrain”. The questionnaire was deployed and administered using the online Survey Monkey platform. The response rate for the online questionnaire was 49.0%.

#### 2.3.1. Dietary Intake Assessment

Dietary intake assessment was performed using a validated food frequency questionnaire (FFQ) [15], consisting of 31 food items that investigated food intake during the past year as a reference period. The responses included six options (never/rarely, once a day, 2 times/day, 1–2 times/week, 2–3 times/week, and 3–6 times/week). Food items listed were grouped into several food groups on the basis of similarity of type of food and nutrient composition [16].

#### 2.3.2. Ultrasound Examination

All patients underwent a standardized ultrasound examination using high-end ultrasound equipment (either Siemens Sonoline Elegra, Siemens ACUSON Sequoia 512 (Siemens, Erlangen, Germany), GE Healthcare Logic 9 (GE Medical Systems, Pewaukee, WI, USA) or Hitachi EUB-8500 (Hitachi Medical Corporation, Tokyo, Japan). The examination of all visible liver parenchyma was performed with a 3–5 MHz transducer. Liver parenchyma was examined with sagittal as well as longitudinal guidance of the probe and completed by lateral and intercostals views. The “abdominal general” setting was used on the US system for all the US examinations. Three radiologists with 5–20 years of experience and more than 600 liver US examinations per year, performed and interpreted the US examination of all participants. All three radiologists were blinded for the data of this study. The following acknowledged scoring items were used: (1) echogenicity of liver parenchyma; (2) visualization of the diaphragm; (3) visualization of intrahepatic vessels; and (4) visualization of posterior part of the right hepatic lobe. Subsequently, the degree of liver steatosis was scored

#### 2.3.3. Demographic, Anthropometric, and Biochemical Assessment

Demographic data such as age, gender, and education level were collected. We also collected information on lifestyle factors (i.e., smoking status, alcohol consumption and sleep duration) using the self-administered questionnaire. The short version of the International Physical Activity Questionnaire (IPAQ) was adopted for physical activity assessment among the categorized MetS patients [17]. According to the total metabolic equivalent task (METs) min per week, three categories of physical activity (low, moderate, and high) were assigned.

Anthropometric measurements of weight and height were taken. Weight was measured with light clothing without shoes by a digital scale to the nearest of 0.1 kg and height was measured using a stadiometer to the nearest of 0.5 cm. Body mass index (BMI) was calculated as weight (kg) divided by height in meters squared (m^2^).

Systolic and diastolic blood pressure measurements were taken 5 min after rest using a standard mercury sphygmomanometer. Biochemical data was obtained from fasting (12 h) blood samples collected from the subjects by venipuncture for the determination of plasma concentrations of FPG, TG, and HDL-C.

### 2.4. Outcomes

The primary outcome of the study was to assess the association between food consumption and changes in biomarkers related to MetS in Bahraini older adults. Secondary outcomes included the effects (association) between sleep, smoking, physical activity, polycystic ovary syndrome, and fatty liver on MetS factors.

### 2.5. Statistical Analysis

The raw data from the responses of each participant were coded numerically and entered in Statistical Package for the Social Sciences (SPSS) version 21.0 (Chicago, IL, USA) software program for analysis. The Kolmogorov-Smirnov normality test was used to examine whether variables were normally distributed. The variables were not normal distributed. Linear regression was used to assess the relationship between food intake and fatty liver as dependent variable at 3 levels. Statistical tests were accomplished to examine the validity and reliability of the questionnaire. Test-retest reliability, internal reliability, and construct validity of the questionnaire were determined. Overall internal reliability (Cronbach’s alpha = 0.90) and test-retest reliability (0.90) were high. Significant differences (*p* < 0.001) between the scores indicated that the food frequency questionnaire had satisfactory construct validity.

Mean and median were calculated for continuous data, including age and BMI. Frequencies were calculated for categorical data, including gender, education, and fatty liver. The Speraman’s correlation analysis with bootstrap was performed in order to assess the association between metabolic markers linked to the metabolic syndrome and food frequencies. The association between fatty liver and food frequencies was evaluated with the multivariate regression analysis with bootstrapping 1000. A two-tailed *p* < 0.05 was considered statistically significant. Scatter plots were generated in SPSS to indicate the relationship between numerical variables.

## 3. Results

### 3.1. Demographic Characteristics of the Study Population

The descriptive demographic characteristics of the study sample are presented in Table 1. More than half of the sample diagnosed with MetS (58.9%) were female, while 41.1% were male. Moreover, more than a third of the study population (36.4%) had an education level of high school or less. Others had higher education qualifications (postgraduate: 1.4%; graduate: 4.6%; postsecondary: 6.6%), while the other half (51.0%) refrained from disclosing their education level. The median age of participants was 65 ± 10 years, and the BMI was in a situation of overweight or obesity. In one subgroup of 61 patients were assessed the fatty liver, 89% were with non-alcoholic fatty liver disease.

### 3.2. Relationship between Food Frequency Consumption and MetS Risk Factors

Table 2 shows the bootstrap partial correlation analysis. Statistically significant correlations were found between dairy products with low fat and SBP (r = 0.182, *p* < 0.001) and lower BMI (r = −0.195; *p* < −0.01). Higher chicken consumption was associated with reduction in BMI (r = −0.273; *p* < −0.01). A higher consumption of ricotta and cheddar cheese was associated with higher levels of triglycerides (*p* < 0.01). Higher frequent consumption of rice (basmati) was associated with lower glucose levels (r = −0.200; *p* < −0.01)

### 3.3. Relationship between Food Frequency Consumption and Fatty Liver

As shown in Table 3, only the consumption of cream cheese was associated with the higher risk of fatty liver.

Figure 1 shows the median levels of the main metabolic markers by fatty liver group.

No statistically significant associations were recorded, although all metabolic parameters linked with metabolic syndrome were higher in people with fatty liver (apart for diastolic blood pressure).

## 4. Discussion

This study found that a higher intake of cheeses such as ricotta, cheddar and cream cheese was associated with higher level of triglycerides and fatty liver. Secondly, the study showed that higher frequent consumption of rice (basmati) was associated with lower glucose levels. Third, more frequent consumption of chicken was associated with lower body mass index. Regarding the first point, higher consumption of cheese is associated with the decrease in triglycerides, which has been widely studied in the literature. Indeed, as reported by De Goede et al., increased serum TG concentrations have long been associated with a higher risk for CVDs; however, whether they promote CVD or are just a biomarker for risk is still debated. This is the first study to have investigated the impact of full fat cheese in a population with MetS, and this is a new finding to be considered for further observational studies [18].

Another point of reflection was shown by a very interesting meta-analysis performed by Miller et al. in a heathy population. Data showed that TG responses did not differ between groups (high consumption of full fat cheeses and not). The authors concluded that the effects on lipids caused by the cheese diet were partially due to the differences in cholesterol concentration and the polyunsaturated to saturated fatty acid ratio of the diets [19].

Our study found higher consumption of chicken to be associated with a lower BMI. As shown by Williamson et al. in an elderly cohort, the higher chicken consumption had satiating properties that persisted for several hours after the meal, which could have an impact on the management of weight [20]. In order to better understand our results, we can deduce that chicken can indirectly affect the CVD by affecting specific metabolic markers, for example, thereby leading an improvement of body weight.

Last but not least, this study showed an important association between basmati rice consumption and glycemia level. In support of our findings, a very interesting study by Seidelmann et al. indeed demonstrated that during a median follow-up of 25 years, an energy range of between 50 and 55% from carbohydrate was associated with the lowest risk of mortality, and both low carbohydrate consumption (<40%) and high carbohydrate consumption (>70%) conferred greater mortality risk than did moderate intake, which was consistent with a U-shaped association leading from 1.23, to 1.11, and to 1.36 for high carbohydrate consumption [21].

Our data contradict those presented in a meta-analysis that showed that higher consumption of white rice was associated with a significantly higher risk of T2D, especially in Asian (Chinese and Japanese) populations. [22]. One possible explanation could be that there are three main factors that appear to explain most of the variation in glycemic and insulinemic responses to rice: (1) inherent starch characteristics (amylose to amylopectin ratio and rice cultivar); (2) post-harvest processing (particularly parboiling); and (3) consumer processing (cooking, storage and reheating) [23].

The population in Bahrain, for cultural reasons, is used to consuming full fat dairy products. The pool of fatty acids in low fat dairy products consumed in Bahrain is not rich in saturated fatty acids. The association between obesity and MetS is well established [24]. The higher prevalence of obesity in Bahrain parallels the trend in Western countries. According to the Bahrain National Health Survey of 2018, 42.8% of the Bahraini population were obese, 40.4% suffered from hypertension, 18.4% from diabetes, 29.4% from high cholesterol, 61.4% had low HDL-C, 20.9% had high low-density lipoprotein-cholesterol, and 39.4% had elevated TGs, all of which contribute to MetS development [25]. As our results show, the average BMI among older Bahraini adults with MetS was 33.49 kg/m^2^, which is in the obese weight status range. In view of the findings of the present study, there is a strong need for implementing effective programs at the national level to improve the awareness, prevention, and treatment of MetS and its related conditions. The implications of this study for future research suggest a need for more investigations that focus on high-quality cohort studies and clinical trials to support and confirm our findings. As a final point, this study did not address the association of fatty liver with alcohol consumption since, none of patients declared a usual use or abuse of alcoholic consumption (less than 1%). It is difficult to quantify the real effect of the alcohol consumption in this setting, because of cultural reasons. As a limitation, this study did not assess the association between metabolic risk factors and lifestyle factors, such as sleep, smoking, and physical activity, as the majority of the participants were sedentary with low physical activity levels. In addition, we calculated many *p*-values, and only a few were significant, so we recognize that they may have occurred by chance. Further research will be needed to confirm the patterns that we found.

This study has several strengths. A major strength is the provision of unique and valuable information and contributing to the data related to metabolic syndrome from Bahrain, as no previous studies have been performed focusing on dietary patterns and the correlation between biochemical biomarkers and MetS in Bahrain. The remarkable feature of our study is the use of a comprehensive approach in collecting data. The sample size is one of the limitations, and the low response rate to the online questionnaire is considered an additional limitation. A self-report questionnaire method was used to examine the food intake frequency, as well as an international physical activity questionnaire, which may have some recall bias. The study has a cross-sectional design, which prevents assessing causality. Additionally, the impact of dietary patterns on MetS is not fully understood and it is possible that MetS diagnosis may influence and change the dietary patterns of the patients after diagnosing. Moreover, some data, like waist circumference, were not collected due to time constraints and improper circumstances.

## 5. Conclusions

In older Bahraini adults with MetS, higher consumption of cheese is linked with a derangement of lipid profile and non-alcoholic fatty liver. Positive effects on BMI were recorded with higher consumption of basmati rice and chicken. Further larger studies are needed to assess metabolic risk factors for MetS.

## Figures and Tables

**Figure 1 geriatrics-07-00075-f001:**
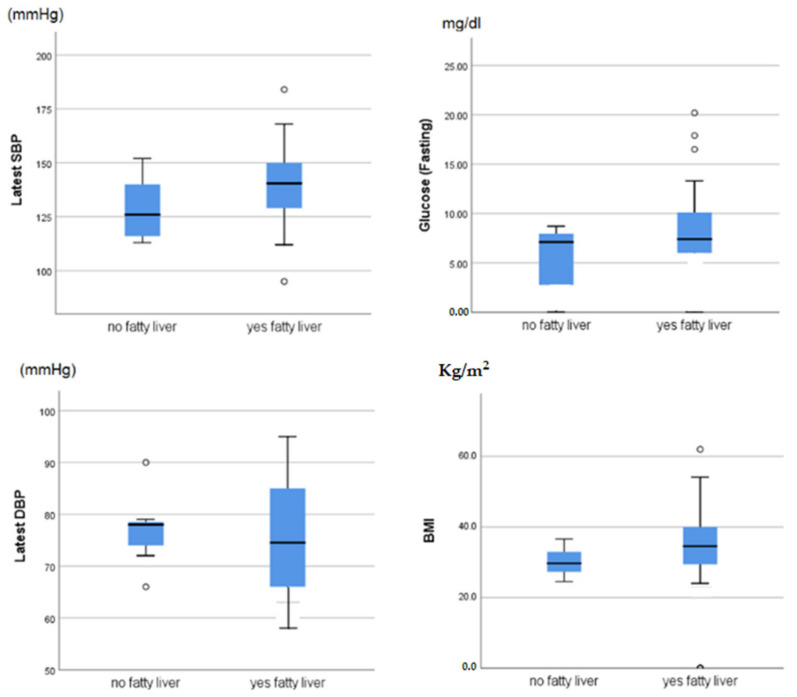
Association between fatty liver and parameters linked to metabolic syndrome.

**Table 1 geriatrics-07-00075-t001:** The descriptive demographic characteristics of Bahraini population with MS.

Variable		(Median IQR)
Age (Years)		65 ± 10
BMI (kg/m)		33.49 ± 6.11
		**(*n*; %)**
Gender	Male	62 (41.1)
	Female	89 (58.9)
Education	Below Secondary School Certificate	26 (17.2)
	Secondary School Certificate	29 (19.2)
	Diploma Degree	10 (6.6)
	Bachelor’s Degree (BSc)	7 (4.6)
	Master’s Degree (MSc)	1 (0.7)
	Doctorate (PhD)	1 (0.7)
	Non-respondent	77 (51)
Metabolic Syndrome Family History	Yes	12 (7.9)
	No	37 (24.5)
	Do not know	25 (16.6)
	Non-respondent	77 (51)
Fatty liver	Yes	54 (89)
	No	7 (11)

**Table 2 geriatrics-07-00075-t002:** Spearman’s partial correlation analysis with bootstrap.

	Latest SBP	Latest DBP	BMI	HDL	Triglycerides	Glucose
Fish	0.071	−0.056	−0.135	0.052	0.014	0.126
Chicken	0.031	0.087	**−0.273 ****	−0.120	0.043	−0.059
Processed Meat	0.075	0.064	−0.124	−0.140	−0.007	0.006
Beef	−0.036	−0.195	−0.072	0.122	−0.013	0.069
Lamb	0.097	−0.061	−0.016	0.087	0.092	0.052
Legumes	0.023	0.019	−0.154	−0.066	0.039	−0.071
Eggs	−0.018	0.074	−0.130	−0.182	0.121	0.041
Rice	−0.035	−0.112	−0.078	0.000	−0.037	**−0.199 ***
Dairy products (Full fat)	−0.013	0.112	−0.012	−0.016	0.050	−0.053
Dairy products (Low fat)	**0.182 ***	0.060	**−0.195 ***	0.024	0.110	0.045
Ricotta or goat cheese	−0.171	−0.060	0.140	−0.021	**0.206 ***	0.058
Cheddar cheese	0.009	−0.024	−0.097	−0.175	**0.195 ***	0.099
Cream cheese	0.022	−0.135	−0.109	0.007	−0.032	−0.051
Butter/Mayonnaise	0.026	−0.043	0.076	−0.109	0.032	0.132
Vegetable oil	−0.072	−0.046	−0.038	−0.025	−0.037	0.042
Olives	0.055	−0.046	−0.130	−0.014	0.019	0.016
Fruits	0.098	−0.003	−0.102	−0.026	0.011	0.064
Vegetables	0.189	−0.008	−0.054	0.090	−0.061	−0.020
Nuts and dried fruits	0.005	−0.066	−0.027	−0.063	−0.047	0.063
Dates	0.149	0.086	−0.098	0.105	0.059	−0.123
Desserts	0.025	0.008	−0.100	−0.090	0.155	0.040
Traditional sweets:	0.135	0.029	0.006	0.064	0.029	0.052
Chips	0.043	−0.060	0.025	0.026	−0.059	0.058
Hamburgers	−0.017	0.039	−0.084	−0.052	−0.003	0.071
Shawarma	−0.025	0.036	−0.169	−0.170	−0.003	0.053
Pizzas and pies	0.073	0.085	−0.006	−0.009	0.145	0.108
Falafel sandwiches	0.129	0.089	−0.103	−0.099	0.126	0.010
French fries	0.004	0.136	0.020	−0.014	0.022	0.127
Sambosa	0.098	0.184	−0.122	−0.002	0.017	0.051
Carbonated beverages	0.125	0.020	0.017	−0.048	−0.030	0.011
Alcoholic drinks	0.030	−0.101	0.151	0.111	0.005	−0.091
Hot drinks: tea, coffee	−0.055	0.116	0.035	−0.129	−0.079	−0.054

* statistically significant at *p* < 0.05. ** statistically significant at *p* < 0.01.

**Table 3 geriatrics-07-00075-t003:** Multivariate regression shows the association between risk of fatty liver and specific foods intake (fatty liver as dependent variable categorized in 3 levels = 0 no fatty liver; 1 not defined; 2 = yes fatty liver).

Variable	βeta	Bias	*p* Value
Fish	0.84	7.648 ^b^	0.298
Chicken	−53.68	42.085 ^b^	0.484
Processed Meat	−154.07	163.977 ^b^	0.405
Beef	203.89	−233.279 ^b^	0.915
Lamb	−167.08	156.712 ^b^	0.956
Legumes lentil peas beans	207.25	−210.479 ^b^	0.304
Eggs	−74.03	69.987 ^b^	0.147
Rice	−28.87	36.894 ^b^	0.460
Dairy products Full fat	−35.00	25.043 ^b^	0.179
Dairy products Low fat	−43.46	57.801 ^b^	0.247
Ricotta or goat cheese	−70.95	72.593 ^b^	0.270
Cheddar cheese	98.19	−97.082 ^b^	0.617
Cream cheese	41.53	−19.240 ^b^	**0.034 ****
Butter Mayonnaise	−42.76	45.696 ^b^	0.241
Vegetable oil	70.56	−73.003 ^b^	0.753
Olives	−33.16	36.963 ^b^	0.747
Fruits	−59.59	49.281 ^b^	0.549
Vegetables	100.52	−118.915 ^b^	0.956
Nuts	83.13	−60.547 ^b^	0.898
Dates	−92.48	91.305 ^b^	0.102
Dessert	−7.73	4.916 ^b^	0.553
Traditional sweets	−247.35	247.516 ^b^	0.935
Chips	−79.85	78.675 ^b^	0.651
Hamburgers	10.69	−11.075 ^b^	0.175
Shawarma	82.29	−82.093 ^b^	0.132
Pizzas and pies	192.56	−192.560 ^b^	0.701
Falafels	1.82	−1.820 ^b^	0.753
French fries	−3.74	3.741 ^b^	0.343
Sambosa	42.10	−42.099 ^b^	1.000
Carbonated beverages and juices	3.80	−3.802 ^b^	0.678
Alcoholic drinks	−46.00	46.002 ^b^	0.701

** statistically significant at *p* < 0.05. ^b^ Bias

## Data Availability

Not applicable.

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
