# Peer review of "Cheese Intake Exhibits an Alteration of Glycolipid Profile and Impacts on Non-Alcoholic Fatty Liver in Bahraini Older Adults"

_geriatrics, 2022, doi:10.3390/geriatrics7040075_

Round 1
Reviewer 1 Report
The authors have revised their manuscript in accordance to the reviewer's suggestions.
As final corrections, I ask to change the wording of "increase" or "decrease" to "lower" or "higher" throughout abstract, results and discussion, as this is a cross-sectional study that does not show changes but differences.
Also, all correlation coefficients should be given with "0." before the decimals in order to provide a consistent format.
Author Response
Thanks a lot for your support and final reading of the manuscript. We did improve the manuscript a lot following your comments.
Regarding your final questions:
Question: As final corrections, I ask to change the wording of "increase" or "decrease" to "lower" or "higher" throughout abstract, results and discussion, as this is a cross-sectional study that does not show changes but differences.
Answer: We replaced all increase with higher and decrease with lower in all manuscript
Question: Also, all correlation coefficients should be given with "0." before the decimals in order to provide a consistent format.
Answer We added all 0 ahead the decimals.
Reviewer 2 Report
The few signficant p-values included in the paper don't look like more than would be expected by chance. In the absence of any effects, 5% of p-values will be <0.05. The authors need to include something about this in the paper. Something along the lines "We have calculated many p-values and only a few are significant, so we recognise that they may have occurred by chance. Further research will be needed to confirm the patterns we have found."
The Methods still states, line 206, that odds ratios were calculated, but these do not appear in the paper.
Author Response
Thanks a lot for reviewing the manuscript. Thanks to your comments we improved a lot this article.
here below you find the final 2 answers to your comments.
QUESTION 1 the few signficant p-values included in the paper don't look like more than would be expected by chance. In the absence of any effects, 5% of p-values will be <0.05. The authors need to include something about this in the paper. Something along the lines "We have calculated many p-values and only a few are significant, so we recognise that they may have occurred by chance. Further research will be needed to confirm the patterns we have found."
Answer: thanks for your suggestion, we included this statement into the discussion as limitation of the study.
QUESTION 2 The Methods still states, line 206, that odds ratios were calculated, but these do not appear in the paper.
Answer: Thanks a lot. We carefully fixed this issue.
This manuscript is a resubmission of an earlier submission. The following is a list of the peer review reports and author responses from that submission.
Round 1
Reviewer 1 Report
The modifications made are correct.
Reviewer 2 Report
The authors submitted the twice-revised version of their manuscript, containing a cross-sectional analysis on the association between dietary factors and major cardiometabolic risk factor outcomes.
The overall rationale of the paper is clear; the specific focus on Middle Eastern cohorts is shown. The scientific background warrants some balance towards null results in all trials, fading associations in later studies, or missing replication in RCTs, for example for saturated fat (Neuenschwander et al. 2018), red meat (Zeraatkar et al. 2019), coffee and so on.
Methods: The chapter on liver fat measurements seems to be incomplete. For statistics, a specific statistician reviewer is recommended. This is of utmost relevance, as data in a previous version of that paper showed, that the association between carbohydrate intake and metabolic outcomes was in fact U-shaped, but not linear. Also, some correction for multiple testing seems necessary.
Results:
Figure 1: Please add scientific units for all axis legends and use adequate decimals for all numbers in the diagrams.
Figure 1: Values for BMI and glucose are implausible, as outliers and whiskers indicate the inclusion of values way below the physiologic range. The statement in line 229/230 is untrue, as DBP is numerically lower in NAFLD cases.
Discussion: Line 250 is misleading. High intake of chicken is associated with lower BMI.
The entire discussion and conclusion does not reflect on the found associations between low-fat dairy, BMI and SBP.
Throughout the manuscript, "decrease/increase" or "reduction" should not be used given the cross-sectional (not longitudinal) nature of the data set.
Reviewer 3 Report
Statistical tests were accomplished to examine the validity and reliability of the 182 questionnaire." This needs more explanation.
The authors state that "The variables were normal distributed". It is surprising that all variables are normally distributed, but if so why were Spearman rather than standard
Pearson correlations calculated?
For assessing the risk (OR) of fatty liver, 89 / 11 patients is not a good sample size. This is less a problem with correlations. Where are OR estimates presented in the paper?
Was any adjustment for total food intake made before the correlations with specific items were examined?
How many correlations are there in Table 2, and how many are significant? We can expect 5% by chance. It doesn't seem to be any more than that. What's seen here is not sufficient to conclude that these correlations are real effects.
Table 3 shows only one significant coefficient out of about 20, which is what would be expected by chance in the absence of any effects. Any multiple testing adjustment would show this. The conclusion should be that there is no evidence of any link with fatty liver, which is to be expected given the very small sample size.